# Incidence of Capillary Leak Syndrome as an Adverse Effect of Drugs in Cancer Patients: A Systematic Review and Meta-Analysis

**DOI:** 10.3390/jcm8020143

**Published:** 2019-01-26

**Authors:** Gwang Hun Jeong, Keum Hwa Lee, I Re Lee, Ji Hyun Oh, Dong Wook Kim, Jae Won Shin, Andreas Kronbichler, Michael Eisenhut, Hans J. van der Vliet, Omar Abdel-Rahman, Brendon Stubbs, Marco Solmi, Nicola Veronese, Elena Dragioti, Ai Koyanagi, Joaquim Radua, Jae Il Shin

**Affiliations:** 1College of Medicine, Gyeongsang National University, Jinju 52727, Korea; pearlmed15@gmail.com; 2Department of Pediatrics, Yonsei University College of Medicine, Yonsei-ro 50, Seodaemun-gu, C.P.O., Box 8044, Seoul 03722, Korea; AZSAGM@yuhs.ac (K.H.L.); jireh18@hanmail.net (I.R.L.); inflames132@naver.com (D.W.K.); aguilera83@naver.com (J.W.S.); 3Division of Pediatric Nephrology, Severance Children’s Hospital, Seoul 03722, Korea; 4Institute of Kidney Disease Research, Yonsei University College of Medicine, Seoul 03722, Korea; 5Wonkwang University School of Medicine, Iksan 54538, Korea; chamomilez@nate.com; 6Department of Internal Medicine IV (Nephrology and Hypertension), Medical University Innsbruck, 6020 Innsbruck, Austria; andreas.kronbichler@i-med.ac.at; 7Luton& Dunstable University Hospital NHS Foundation Trust, Lewsey Road, Luton LU4 ODZ, UK; michael_eisenhut@yahoo.com; 8Department of Medical Oncology, Amsterdam UMC, Cancer Center Amsterdam, VU University, 1081 HV Amsterdam, The Netherlands; JJ.vanderVliet@vumc.nl; 9Clinical Oncology Department, Faculty of Medicine, Ain Shams University, Cairo 11566, Egypt; omar.abdelsalam@ahs.ca; 10Department of Oncology, University of Calgary, Tom Baker Cancer Centre, Calgary, AB T2N 1N4, Canada; 11Institute of Psychiatry, Psychology and Neuroscience, King’s College London, De Crespigny Park, London SE5 8AF, UK; brendon.stubbs@kcl.ac.uk; 12South London and Maudsley NHS Foundation Trust, Denmark Hill, London SE5 8AZ, UK; 13Faculty of Health, Social Care and Education, Anglia Ruskin University, Chelmsford CM1 1SQ, UK; 14Department of Neuroscience, University of Padova, 35121 Padova, Italy; marco.solmi83@gmail.com; 15National Research Council, Neuroscience Institute, 35128 Padova, Italy; ilmannato@gmail.com; 16Pain and Rehabilitation Centre, and Department of Medical and Health Sciences, Linköping University, SE-581 85 Linköping, Sweden; elena.dragioti@liu.se; 17Parc Sanitari Sant Joan de Déu/CIBERSAM, Universitat de Barcelona, Fundació Sant Joan de Déu, Sant Boi de Llobregat, 08830 Barcelona, Spain; a.koyanagi@pssjd.org; 18Institut d’Investigacions Biomèdiques August Pi i Sunyer (IDIBAPS), 08036 Barcelona, Spain; radua@clinic.cat; 19Mental Health Research Networking Center (CIBERSAM), 08036 Barcelona, Spain; 20Department of Psychosis Studies, Institute of Psychiatry, Psychology and Neuroscience, King’s College London, London SE5 8AF, UK; 21Centre for Psychiatric Research, Department of Clinical Neuroscience, Karolinska Institutet, 113 30 Stockholm, Sweden

**Keywords:** capillary leak syndrome, cancer, interleukin-2, anti-CD agents, bone marrow transplantation

## Abstract

Capillary leak syndrome (CLS) is a rare disease with profound vascular leakage, which can be associated with a high mortality. There have been several reports on CLS as an adverse effect of anti-cancer agents and therapy, but the incidence of CLS according to the kinds of anti-cancer drugs has not been systemically evaluated. Thus, the aim of our study was to comprehensively meta-analyze the incidence of CLS by different types of cancer treatment or after bone marrow transplantation (BMT). We searched the literatures (inception to July 2018) and among 4612 articles, 62 clinical trials (studies) were eligible. We extracted the number of patients with CLS, total cancer patients, name of therapeutic agent and dose, and type of cancer. We performed a meta-analysis to estimate the summary effects with 95% confidence interval and between-study heterogeneity. The reported incidence of CLS was categorized by causative drugs and BMT. The largest number of studies reported on CLS incidence during interleukin-2 (IL-2) treatment (*n* = 18), which yielded a pooled incidence of 34.7% by overall estimation and 43.9% by meta-analysis. The second largest number of studies reported on anti-cluster of differentiation (anti-CD) agents (*n* = 13) (incidence of 33.9% by overall estimation and 35.6% by meta-analysis) or undergoing BMT (*n* = 7 (21.1% by overall estimation and 21.7% by meta-analysis). Also, anti-cancer agents, including IL-2 + imatinib mesylate (three studies) and anti-CD22 monoclinal antibodies (mAb) (four studies), showed a dose-dependent increase in the incidence of CLS. Our study is the first to provide an informative overview on the incidence rate of reported CLS patients as an adverse event of anti-cancer treatment. This meta-analysis can lead to a better understanding of CLS and assist physicians in identifying the presence of CLS early in the disease course to improve the outcome and optimize management.

## 1. Introduction

Capillary leak syndrome (CLS), also known as vascular leak syndrome (VLS), is a rare but fatal disease, and an idiopathic form of CLS was first reported by Clarkson in 1960 [1]. Patients with CLS show a profound increase of capillary permeability, which can result in the leakage of plasma with proteins out from capillaries, resulting in clinical features such as edema, hypotension, hypoalbuminemia, or hemoconcentration [2,3,4]. Most cases of CLS are classified as idiopathic forms, and its pathogenesis has not been elucidated yet. It may also develop as a secondary form, preceded by autoimmune diseases, infections, snakebites, and drugs [5]. Cancer and chemotherapy are also considered to be important causes of secondary CLS, but the underlying mechanisms remain mostly elusive [4,6]. CLS shows a high mortality rate, with one-year and five-year survival rates being 89% and 73%, respectively, in idiopathic forms [7]. If prophylactic treatment including intravenous immunoglobulin (IVIG) is provided, disease-specific mortality seems to decrease in idiopathic CLS [3,7,8]. However, there is no established treatment for secondary CLS, and supportive therapy with fluid management may be the most important element [5]. Currently, exact treatment guidelines for CLS do not exist [8,9,10,11]. Moreover, the capillary leak phenomenon can be similar between idiopathic CLS and secondary forms of CLS due to drugs, but the pathophysiology of them may be somewhat different.

CLS has also been reported as an adverse event in cancer patients receiving different types of anti-cancer treatments [3]. However, there has been a lack of awareness of CLS by oncologists due to the non-specific symptoms of this disease [3], and the incidence of CLS according to the different types of anti-cancer agents or therapy has not been systematically investigated. 

Thus, in this study, we conducted a systematic review and meta-analysis to estimate the incidence proportion of CLS in cancer patients who received specific anti-cancer treatment or therapy, including bone marrow transplantation (BMT). 

## 2. Methods

### 2.1. Literature Search Strategy and Study Selection

We followed the guideline of Preferred Reporting Items for Systematic Reviews and Meta-Analyses (PRISMA) checklist for this systematic review (Appendix A). Two investigators (K.H.L. and I.R.L.) manually searched the literature (PubMed and EMBASE) to find original studies that reported cases of CLS as an adverse event in cancer patients who received specific cancer treatment or BMT. The search terms were: “(Capillary leak OR Vascular leak) AND (cancer OR carcinoma OR neoplasm OR tumor)”, and the date of the last search was 15 July 2018. If there was a discrepancy for the inclusion/exclusion of the respective article, it was discussed and resolved by consensus among three investigators (J.I.S., K.H.L., and I.R.L.). The full literature search strategy is presented in Figure 1.

The eligibility criteria for inclusion were: studies on (1) CLS that was an adverse event of cancer treatment-related drugs; and (2) CLS that developed after BMT; and the exclusion criteria were: studies on (1) CLS that were caused by idiopathic forms, infection, or surgery; and (2) CLS attributed to cancer itself, or (3) missing raw data from the original study reporting on CLS as an adverse event of cancer treatment. Our initial search yielded 4612 articles, but we finally included 62 clinical trials (or studies) that met the inclusion criteria for this systematic review.

### 2.2. Data Extraction

For each eligible clinical trial (or study), we recorded the first author, publication year, journal name, period of study, country, total number of patients, number of patients who developed CLS, diagnosed cancer type, causative drugs, and the dose of drugs. 

### 2.3. Analyses of Clinical Trials (or Studies)

The incidence of CLS for each study was estimated by calculating the ratio between the number of CLS patients and the total number of cancer patients who received the causative drug or BMT. The data for each study are presented in Table 1 [12,13,14,15,16,17,18,19,20,21,22,23,24,25,26,27,28,29,30,31,32,33,34,35,36,37,38,39,40,41,42,43,44,45,46,47,48,49,50,51,52,53,54,55,56,57,58,59,60,61,62,63,64,65,66,67,68,69,70,71,72,73]. To estimate the incidence of CLS for the relevant groups, we presented the data with median (ranges) and also performed a meta-analysis to estimate the summary effects with proportion of CLS and 95% confidence interval (CI) using random-effect models [74,75]. Random effects meta-analysis provides the weighted average of the effect sizes of a group of studies with the assumption that individual studies are estimating different effects [76]. We evaluated the between-study heterogeneity using the *I*^2^ metric of inconsistency and P value of the χ^2^-based Cochran Q test. *I*^2^ is the ratio of the between-study variance over the sum of the within-study and between-study variances, and it ranges between 0–100%. *I*^2^ values of <25%, 25–50%, and >75% are usually judged to represent low, moderate (large), and high (very large) heterogeneity, respectively [77]. Since statistical tests for heterogeneity are not very powerful, a higher *p* value than usual (*p* < 0.10: significant heterogeneity) is used as the cut-off for clinical heterogeneity [78].

### 2.4. Statistical Analysis

To meta-analyze the incidence of CLS according to the causative anti-cancer drugs or after BMT, the summary effects with 95% CI and the between-study heterogeneity were analyzed by using MedCalc version 15.8 software (MedCalc Software, Ostend, Belgium).

## 3. Results

There were 62 clinical trials (or studies) that reported on the incidence of CLS in patients receiving anti-cancer treatments or after BMT. Most of these studies were clinical trials in which the incidence of CLS was reported as an adverse event of anti-cancer treatment (Table 1) [12,13,14,15,16,17,18,19,20,21,22,23,24,25,26,27,28,29,30,31,32,33,34,35,36,37,38,39,40,41,42,43,44,45,46,47,48,49,50,51,52,53,54,55,56,57,58,59,60,61,62,63,64,65,66,67,68,69,70,71,72,73]. Among these, six studies reported on CLS associated with BMT with or without other agents (Table 2) [40,42,70,71,72,73]. The results of meta-analyses on the incidence of CLS induced by various drugs in cancer patients are summarized in Table 3 and Appendix A.

There were 18 studies that reported on the incidence of CLS associated with the use of interleukin-2 (IL-2), which ranged from 5.3% to 100%. The incidence of CLS by IL-2 was 34.7% by overall estimation and 43.9% by meta-analysis. Although varying treatment doses were used, no correlations were found between the dose of IL-2 and the overall incidence of CLS. IL-2 was used in combination with other agents in several studies. These included combinations with bevacizumab (one study), imatinib mesylate (one study, three dose-related results), taurolidine (one study), interferon (IFN)-alpha (two studies), chimeric human/murine anti-GD2 ch14.18 monoclonal antibody (mAb) (one study), granulocyte-macrophage colony-stimulating factor (GM-CSF) + granulocyte colony-stimulating factor (G-CSF) (one study), GM-CSF + anti-GD2 mAb + isotretinoin (one study) and 5-fluorouracil (5-FU) (two studies). The incidence of CLS in patients treated with IL-2 with other agents was 29.1% by overall estimation and 32.0% by meta-analysis. We found that the highest incidence of CLS (80.5% and 100%) was observed when IL-2 was combined with IFN-alpha. In the IL-2 + imatinib mesylate group, there was a dose-related increase in the incidence of CLS (0% → 9% → 33.3%). The incidence of CLS in patients who received IL-2 + bevacizumab (IL-2 dose: 9 μg/kg) was 100%. In cases with concomitant IL-2 + 5-FU treatment, the incidence of CLS varied from 6.3% to 25.0%, resulting in 17.5% by overall estimation and 17.1% by meta-analysis.

Two studies reported on the incidence of CLS associated with the use of IL-1 in combination with carboplatin (one study, 40% CLS incidence) or etoposide (one study, 44.4%). Three studies reported on the incidence of CLS associated with the use of GM-CSF, which ranged from 6.8% to 15.0%. The incidence of CLS in patients treated with GM-CSF was low (9.0%) by overall estimation and 10.1% by meta-analysis. The incidence of CLS by GM-CSF was 9.0% by overall estimation and 10.1% by meta-analysis.

Three studies reported on the incidence of CLS associated with the use of gemcitabine, which was very low (2.8–4.3%). The incidence of CLS caused by gemcitabine was 3.5% by overall estimation and 4.9% by meta-analysis. There were two studies that reported on the incidence of CLS associated with the use of SS1P (recombinant anti-mesothelin immunotoxin), which was 5.9% and 54.2%, and showed no dose-response. 

Thirteen studies reported on the incidence of CLS associated with the use of various kinds of anti-cluster of differentiation (CD) agents, which ranged from 5.9% to 100%. The incidence of CLS by various kinds of anti-CD agents was 33.9% by overall estimation and 35.6% by meta-analysis. There were four studies that reported on the incidence of CLS associated with the use of anti-CD22 mAb, which ranged from 11.5% to 100%. The incidence of CLS by various kinds of anti-CD22 mAb was 40.7% by overall estimation and 48.1% by meta-analysis. It appeared that there was an increasing incidence of CLS with an increasing treatment dose of anti-CD22 mAb. The addition of anti-CD19 mAb to anti-CD22 mAb treatment did not result in a further increase in the incidence of CLS. Three studies reported on the incidence of CLS associated with the use of anti-CD25, which ranged from 6.7% to 100%. The incidence of CLS by various kinds of anti-CD25 was 36.7% by overall estimation and 42.2% by meta-analysis.

There were single studies reporting on other drugs associated with CLS in cancer patients, with an incidence of CLS ranging from 3.4% to 80%. The incidence of CLS was high with the use of pyrrolobenzodiazepine (one study, 62.5%), paclitaxel (one study, 80.0%), and moderate with the use of anti-B4-bR (B-cell restricted immunotoxin anti-B4-blocked ricin) (one study, 41.7%), FK973 (novel, substituted dihydro benzoxazine structurally similar to mitomycin) (one study, 35.3%), and low with the use of SGN-10 (a single-chain immunotoxin) (one study, 2.2%), clofarabine + cytarabine + liposomal daunorubicin (one study, 3.4%), cyclosporine (one study, 8.3%), dihydro benzoxazine (one study, 11.1%), ABR-217620 (naptumomab estafenatox) (one study, 15.4%), and ricin A chain-containing immunotoxin (one study, 21.4%).

There were seven studies reporting the incidence of CLS associated with BMT with or without other agents, which ranged from 6.8% to 52.7%. The incidence of CLS associated with BMT was 21.1% by overall estimation and 21.7% by meta-analysis (Table 2 and Table 3, Appendix A).

## 4. Discussion

CLS is an important medical condition that is characterized by the escape of blood plasma into the interstitial space, resulting in edema, hypoalbuminemia, hemoconcentration, and low blood pressure [2]. The pathogenesis of secondary CLS due to anti-cancer treatment is not well-known, but there are several studies supporting the role of pathogenic molecules of idiopathic CLS including multiple cytokines, angiopoietin-2, and vascular endothelial growth factor (VEGF) [5,10,79,80], although the pathophysiology of idiopathic and secondary CLS may be somewhat different, because CLS by anti-cancer drugs could also develop due to a direct toxicity to the capillary system. These molecules are mostly related to an increase in the permeability of vascular endothelial cells leading to vascular leakage. Especially, multiple animal studies suggest that IL-2 causes the acute injury of normal tissues by enhancing neutrophil adhesion and generating reactive oxygen intermediates, proteases, and pro-inflammatory cytokines such as tumor necrosis factor alpha (TNF alpha), which can cause a vascular leakage [81,82]. The proposed pathogenesis of CLS is demonstrated in Figure 2.

Evaluating the incidence of CLS is challenging because the clinical presentations of CLS are non-specific, and it is expected that cases have been misdiagnosed in the past. Recently, CLS has been increasingly diagnosed due to increased awareness of the disease [4]. CLS due to anti-cancer drugs has been sporadically reported in the literature, and it has recently been registered in VigiBase (http://www.vigiaccess.org/), the World Health Organization global Individual Case Safety Report (ICSR) database, which contains reports of suspected adverse drug reactions (ADRs) collected by national drug authorities in more than 130 countries between 1967 and February 2018 [10]. However, it did not report the incidence rate of CLS for patients treated with anti-cancer drugs, and our study firstly reported the incidence of CLS according to the drugs or after BMT by meta-analysis.

Due to the lack of an overall understanding of CLS as an adverse effect of anti-cancer treatment, we carried out a systematic analysis of published clinical trials (or studies) to evaluate the incidence of anti-cancer treatment-related CLS. Through calculating the number of CLS among total patients reported in clinical trials (or studies), we were able to estimate the pooled incidence of CLS when patients were treated with several anti-cancer treatment-related drugs and after BMT. The most studied drug was IL-2, which was used as a cancer immunotherapy, and the incidence of CLS was 34.7% by overall estimation, suggesting that it may be a common adverse effect, and that the phenomenon of CLS has been underestimated in cancer patients in the past. In addition, the incidence of CLS in cancer patients differed according to the specific drug or drug combinations that were used and ranged from 5.3% to 100.0%. Therefore, our analysis shows important results that oncologists should be aware of. However, these studies did not report on the treatment strategies or clinical outcome of CLS, because most studies reported CLS as an adverse event of the drug. The clinical and laboratory data, treatment modalities, and mortality rate of patients and contributing factors leading to mortality of CLS in cancer are well analyzed in our recent systematic review of sporadic case reports [4].

We also found that BMT may be an important risk factor for CLS in cancer patients. The incidence of CLS associated with BMT with or without other agents ranged from 6.8% to 52.7%. The pathophysiology of CLS in BMT-related CLS has not been fully studied, but some hypotheses on the contributing factors have been suggested such as pivotal contribution by circulating leukocytes, decreased C1 esterase inhibitor activity, elevated C4d concentrations, the use of G-CSF or GM-CSF, and elevation of terminal complement complex (TCC) levels [70,71,72,73]. Future studies in this area may shed light on the pathophysiology of CLS associated with BMT and trigger the development of novel therapeutic approaches.

Besides IL-2 and BMT, we identified several potential causative drugs of CLS. The overall estimation of CLS incidence by causative drugs varied from 3.5% (gemcitabine, three studies) to 100% (IL-2 + bevacizumab, one study). Studies with IL-2 + bevacizumab (one study, 100%) and IL-2 + IFN-alpha 2a (two studies, overall estimation 85.5%, meta-analysis 90.4%) showed relatively high CLS incidence proportions, while studies with gemcitabine (three studies, overall estimation 3.5%, meta-analysis 4.9%) and GM-CSF (three studies, overall estimation 9.0%, meta-analysis 10.1%) showed low incidence. Likewise, anti-cancer agents, including IL-2 + imatinib mesylate (three studies) and anti-CD22 mAb (four studies) showed a dose-dependent increase in the incidence of CLS. Considering the small number of studies, it is difficult to state whether there are dose-related trends for these agents. Further studies should be performed to clarify this relationship in order to establish comprehensive therapeutic guidelines, taking CLS as an adverse effect into account. 

Several limitations of this study should be considered. First, the studies that we included were not placebo-controlled trials with a control arm that would allow defining how much of the CLS was attributable to treatment rather than the type and severity of the treated condition. Since the included individual studies just reported the number of CLS as an adverse event of anti-cancer drugs, the prognosis and long-term outcome of CLS could not be addressed. Therefore, further clinical trials or observational studies should attempt to address the prospective associations between CLS and anti-cancer treatment. Second, coexisting conditions were not considered in our study. For example, we extracted the name of the causative drug and its dose, but other effects such as drug combination or cumulative effects may have affected the outcomes. Also, there might be other causes for CLS besides anti-cancer treatment, so potential confounders should be acknowledged.

## 5. Conclusions

Our study is the first systematic analysis of the incidence of CLS in cancer patients treated with various anti-cancer agents and therapy. The incidence of CLS due to IL-2 (18 studies) was 34.7% by overall estimation and 43.9% by meta-analysis, and the corresponding figures for BMT were 21.1% and 21.7%, respectively CLS was also reported in cases receiving other agents. Our study results highlight the need for inclusion of the risk of development of CLS in the choice of treatment and preparation of the appropriate management for cancer patients in anticipation of this syndrome. Thus, we recommend that physicians and oncologists should be aware of secondary CLS in cancer patients during anti-cancer treatment, and encourage careful observation to prevent CLS or enable timely management when CLS develops.

## Figures and Tables

**Figure 1 jcm-08-00143-f001:**
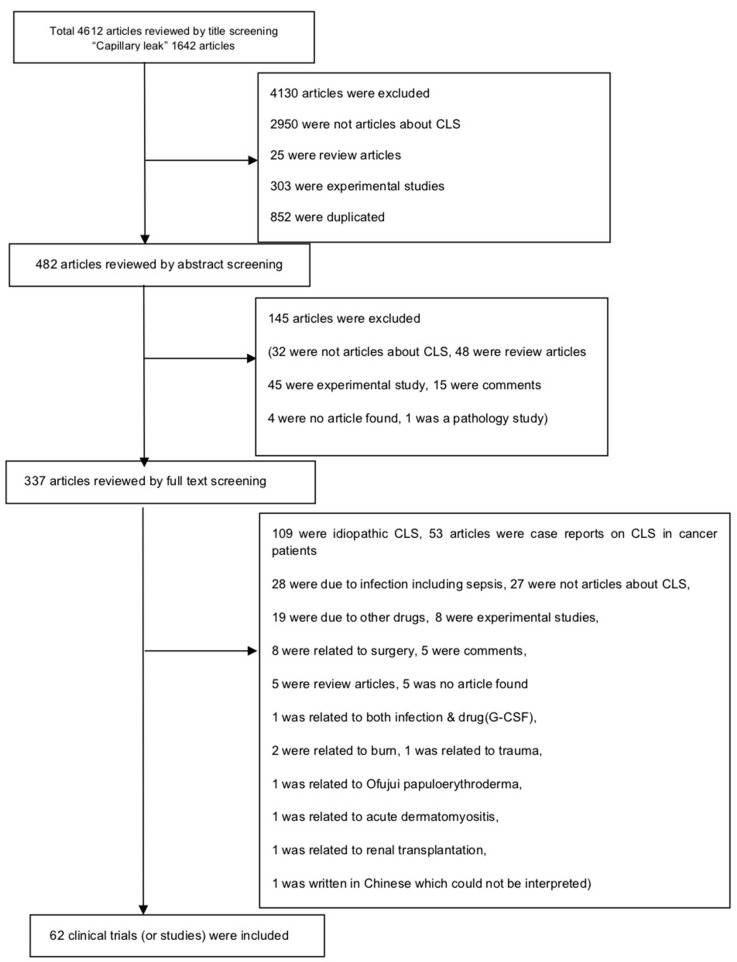
Flow chart of literature search. CLS: Capillary leak syndrome, G-CSF: Granulocyte colony-stimulating factor.

**Figure 2 jcm-08-00143-f002:**
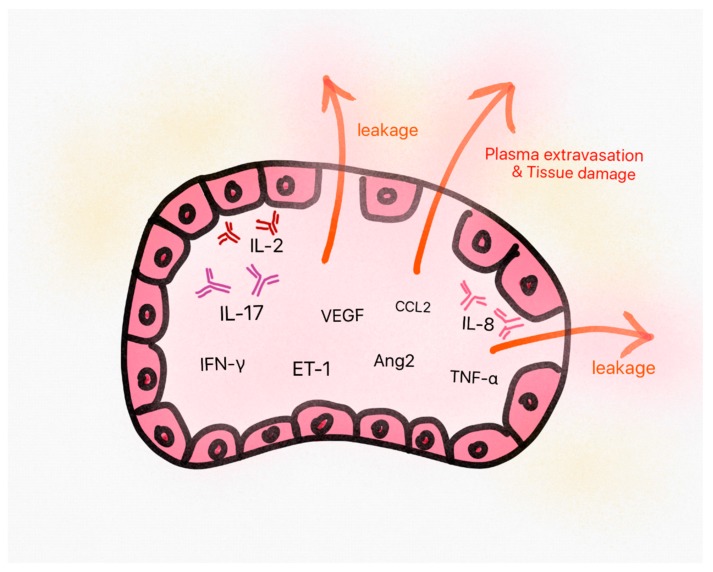
The proposed pathogenesis of capillary leak syndrome (CLS). Some pathogenic molecules in CLS show increased levels in sera, which triggers endothelial cell damage and plasma leakage from vessels. This is supposed to eventually result in the classic triad of symptoms (hypoalbuminemia, hemoconcentration, and hypotension) and normal tissue damages. VEGF: vascular endothelial growth factor, IL: Interleukin, TNF: Tumor necrosis factor, IFN: Interferon, ET: endothelin, CCL: chemokine ligand, Ang2: angiopoietin-2.

**Table 1 jcm-08-00143-t001:** Summary profiles of clinical trials that reported capillary leak syndrome as an adverse event of anti-cancer drugs.

Ref. No.	Author, Year of Publication	Period of Study	Country	Total Number	CLS	Incidence (%)	Diagnosis	Drug	Treatment Dose
**IL-2**									
[12]	Atkins et al., 1999	1985–1993	USA	270	92	34.1	Melanoma	IL-2	720,000 IU/kg every 8 h
[13]	Sparano et al., 1993	1988–1992	USA	44	40	90.9	Melanoma	IL-2	6 × 10^6 ^IU/m^2^ every 8 h
[14]	Tarhini et al., 2007	2000–2003	USA	26	7	26.9	Melanoma	IL-2	600,000 IU/kg every 8 h for up to 14 doses for 2 cycles
[15]	Talpur et al., 2012	2003–2008	USA	8	6	75.0	Cutaneous peripheral T-cell lymphoma	IL-2	Dose level 18 μg/kg
[16]	Gallagher et al, 2007	2006	Israel	14	14*	100.0	Melanoma, renal cell carcinoma	IL-2	Dose level 8–14 μg/kg
[17]	Shusterman et al., 2010	NA	USA	39	12	30.8	Neuroblastoma	IL-2	Dose level 12 mg/m^2^
[18]	Shaughnessy et al., 2005	NA	USA	2	1	50.0	Non-Hodgkin lymphoma, Hodgkin disease, acute leukemia, myelodysplastic syndrome, chronic myelogenous leukemia, multiple myeloma, aplastic anemia	IL-2	Dose level 9.0 μg/kg
[18]	Shaughnessy et al., 2005	NA	USA	20	2	10.0	Non-Hodgkin lymphoma, Hodgkin disease, acute leukemia, myelodysplastic syndrome, chronic myelogenous leukemia, multiple myeloma, aplastic anemia	IL-2	Dose level 4.5 μg/kg
[19]	Frankel et al., 2003	NA	USA	18	2	11.1	Chronic lymphocytic leukemia	IL-2	Dose level 9 or 18 μg/kg
[20]	Duvic et al., 2002	NA	USA	71	18	25.4	Cutaneous T-cell lymphoma	IL-2	Dose level 9 or 18 μg/kg
[21]	Foss et al., 2001	NA	USA	15	2	13.3	Cutaneous T-cell lymphoma	IL-2	Dose level 9 or 18 μg/kg
[22]	Sievers et al., 2000	NA	USA	60	7	11.7	Acute myelogenous leukemia	IL-2	9,000,000 IU/m^2^ for 4 days and 16,000,000 IU/m^2 ^for 10 days
[23]	Duvic et al., 1998	NA	USA	4	1	25.0	Cutaneous T cell lymphoma	IL-2	Dose level 9 or 18 μg/kg
[24]	Meehan et al., 1997	1993–1995	USA	57	3	5.3	Breast cancer	IL-2	MTD 6 × 10^6 ^IU/m^2^/day
[25]	Chang et al., 1993	NA	Japan	20	15	75.0	Melanoma, renal cell cancer	IL-2	Using vaccine-primed lymph node cell with IL-2 (180,000 IU/kg)
[26]	van Haelst Pisani C et al., 1991	NA	France	5	4	80.0	Melanoma, renal cell cancer	IL-2	Human recombinant IL-2 3 × 10⁶ IU/m²/24 h for 4 or 5 days
[27]	Philip et al., 1989	1987–1988	France	20	8	40.0	Renal cell cancer	IL-2	IL-2 3 × 10⁶ IU/m² with lymphapheresis(17), IL-2 3 × 10 ⁶IU/m²(3)
[28]	Carey et al., 1997	NA	UK	10	10*	100.0	Malignant melanoma, renal cell cancer	IL-2	Using 3 × 10⁶ IU/m²/day for 5 days
**IL-2 with other agents**									
[16]	Gallagher et al., 2007	2006	Israel	4	4	100.0	Renal cell carcinoma	IL-2 + bevacizumab	IL-2 dose level 9–14 μg/kg
[29]	Pautier et al., 2013	NA	France	3	0	0.0	Melanoma, ovarian adenocarcinoma, Merkel-cell carcinoma, gastrointestinal stromal tumor, rectal adenocarcinoma, cervical adenocarcinoma	IL-2 + imatinib mesylate	IL-2: 3,000,000 IU/day, imatinib mesylate 400 mg/day
[29]	Pautier et al., 2013	NA	France	11	1	9.0	Melanoma, ovarian adenocarcinoma, Merkel-cell carcinoma, gastrointestinal stromal tumor, rectal adenocarcinoma, cervical adenocarcinoma	IL-2 + imatinib mesylate	IL-2: 6,000,000 IU/day, imatinib mesylate 400 mg/day
[29]	Pautier et al., 2013	NA	France	3	1	33.3	Melanoma, ovarian adenocarcinoma, Merkel-cell carcinoma, gastrointestinal stromal tumor, rectal adenocarcinoma, cervical adenocarcinoma	IL-2 + imatinib mesylate	IL-2: 9,000,000 IU/day, imatinib mesylate 400 mg/day
[30]	O’Brien et al., 2006	NA	Ireland	10	0	0.0	Melanoma	IL-2 + taurolidine	IL-2 72 MIU/m^2^ for 120 h Taurolidine 2% w/v via continuous infusion
[31]	Pichert et al., 1991	1988–1989	Switzerland	14	14*	100.0	Renal cell carcinoma, melanoma	IL-2 + IFN-alfa 2a	IL-2 3 MIU/m^2^ for 4 days IFN-alpha 6 MIU/m^2^ for 2 days (1, 4 day)
[13]	Sparano et al., 1993	1988–1992	USA	41	33	80.5	Melanoma	IL-2 + IFN-alfa	IL-2 4.5 × 10^6 ^IU/m^2^ per dose IFN-alpha 2 4.5 × 10^6 ^IU/m^2^
[32]	Gilman et al., 2009	1997–2002	USA	19	3	15.8	Neuroblastoma	IL-2 + ch14.18	Ch14.18 20 and 40 mg/m^2^/day IL-2 4.5 × 10^6 ^IU/m^2^/day
[33]	Meehan et al., 2010	NA	USA	12	2	16.7	Multiple myeloma, non-Hodgkin lymphoma	IL-2 + GM-CSF + G-CSF	IL-2 6 × 10^5–^1.5 × 10^6 ^IU/m^2^ G-CSF 5 μg/kg GM-CSF 7.5 μg/kg
[34]	Yu et al., 2010	2001–2009	USA	226	51	22.6	Neuroblastoma	IL-2 + GM-CSF + anti-GD2 + isotretionoin	IL-2 3.0 × 10^6 ^IU/m^2^ (week 1), 4.5 × 10^6 ^IU/m^2^ (week 2) GM-CSF 250 μg/m^2^ isotretionoin 160mg/m^2^
[35]	Hamblin et al., 1993	1988–1989	UK	16	1	6.3	Metastatic colorectal cancer	IL-2 + 5-FU	IL-2 18 × 10 IU/m^2^/day over 120 h 5FU 600 mg/m^2^
[36]	Savage et al., 1997	NA	UK	24	6	25.0	Metastatic renal cancer	IL-2 + 5-FU	IL-2 9 × 10^6 ^IU 5-FU 200 mg/m^2^
**IL-1 with other agents**									
[37]	Smith et al., 1993	1990–1992	USA	15	6	40.0	Colon cancer, melanoma, renal cell cancer, lung cancer, pancreatic cancer, liposarcoma, adenocarcinoma with unknown primary site	IL-1 alpha + carboplatin	IL-1 alpha 0.03, 0.1, 0.3 μg/kg carboplatin 800 mg/m^2^
[38]	Worth et al., 1997	1994	USA	9	4	44.4	Osteosarcoma	IL-1 alpha + etoposide	IL-1 alpha 0.1 *μ*g/kg etoposide 100 mg/m^2^
**IL-4**									
[39]	Sosman et al., 1994	NA	USA	17	2	11.8	Renal cell carcinoma, melanoma, colon carcinoma, cholangiocarcinoma	IL-4+IL-2	IL-4 40–600 μg /m^2^/day IL-2 11.2 MIU/m^2^/day
**GM-CSF**									
[40]	Gorin et al., 1992	1988–1990	France	44	3	6.8	Non-Hodgkin lymphoma	GM-CSF	Dose level 250 μg/m^2^
[41]	Liberati et al., 1991	NA	Italy	14	1	7.1	Non-Hodgkin lymphoma	GM-CSF	Dose level 5 μg/kg
[42]	Steward et al., 1989	NA	USA &UK	20	3	15.0	Metastatic solid tumors	GM-CSF	Using dose 0.3, 1.0, 3.0, 10, 30, and 60 μg/kg/day Dose level 32 μg /kg
**Gemcitabine**									
[43]	Jidar et al., 2009	NA	France	23	1	4.3	Cutaneous T-cell lymphoma	Gemcitabine	Using dose 700–1000 mg/m^2^
[44]	Kurosaki et al., 2009	2003–2006	Japan	27	1	3.7	Pancreatic cancer	Gemcitabine	Dose level 1000 mg/m^2^ biweekly
[45]	Dumontet et al., 2001	1988–2000	France	36	1	2.8	Non-Hodgkin lymphoma	Gemcitabine	Dose level 1 g/m^2^
**SS1P**									
[46]	Kreitman et al., 2009	NA	USA	24	13	54.2	Peritoneal mesothelioma, pleural mesothelioma, pleural–peritoneal mesothelioma, ovarian carcinoma, pancreatic carcinoma	SS1P	Dose level 4–25 μg/kg
[47]	Hassan et al., 2007	2000–2006	USA	34	2	5.9	Peritoneal mesothelioma, pleural mesothelioma, pleural–peritoneal mesothelioma, ovarian carcinoma, pancreatic carcinoma	SS1P	Dose level 18 or 25 μg/kg
**Anti-CD agents**									
[48]	Sausville et al., 1995	NA	USA	11	4	36.4	B-cell lymphoma	Anti-CD22	Dose level 28.8 mg/m^2^ MTD 19.2 mg/m^2^
[49]	Vitetta et al., 1991	NA	USA	15	15*	100.0	B-cell lymphoma	Anti-CD22	Using dose 12.5, 25, 50, 75, 100 mg/m^2^
[50]	Wayne et al., 2014	NA	USA	7	2	28.6	Acute lymphoblastic leukemia	Anti-CD22	Dose level 30 μg/kg
[51]	Amlot et al., 1993	NA	USA	26	3	11.5	B-cell lymphoma	Anti-CD22	Using Maximal single dose 2.5–13.9 mg/m^2^
[52]	Stathis et al., 2014	NA	Switzerland	5	1	20.0	Non-Hodgkin lymphoma	Anti-CD22 + temsirolimus	Using dose Anti-CD22 0.8 mg/m^2 ^+ temsirolimus 15 mg/day, Anti-CD22 0.8 mg/m^2^ + temsirolimus 10 mg/day
[53]	Schindler et al., 2011	NA	USA	17	1	5.9	B-cell acute lymphoblastic leukemia	Anti-CD19 + anti-CD22	Dose level 8 mg/m^2^
[54]	Bachanova et al., 2015	NA	USA	25	7	28.0	Pre-B acute lymphoblastic leukemia, chronic lymphocytic leukemia, Non-Hodgkin lymphoma	Anti-CD19 + anti-CD22	Dose level 40–60 μg/kg
[55]	Schnell et al., 2003	NA	Germany	27	3	11.1	Hodgkin lymphoma	Anti-CD25	Dose level 15–20 mg/m^2^
[56]	Schnell et al., 2000	NA	Germany	18	18*	100.0	Hodgkin lymphoma	Anti-CD25	Dose level 15 mg/m^2^/cycle
[57]	Engert et al., 1997	NA	Germany	15	1	6.7	Hodgkin lymphoma	Anti-CD25	Dose level 5 mg/m^2^(3), 10 mg/m^2^(3), 15 mg/m^2^(6), 20 mg/m^2^(3)
[58]	Schnell et al., 2002	NA	Germany	17	3	17.6	Hodgkin lymphoma, Non-Hodgkin lymphoma	Anti-CD30	Dose level 7.5 mg/m^2^(1), 10 mg/m^2^(2) MTD 5 mg/m^2^
[59]	Stone et al., 1996	NA	USA	23	16	69.6	Non-Hodgkin lymphoma	Anti-CD19 + IgG-HD37-dgA	MTD 19.2 mg/m^2^
[60]	Uckun et al., 1999	1996–1998	USA	15	1	6.7	Acute lymphoblastic leukemia, chronic lymphocytic leukemia	CD19 receptor directed tyrosine kinase inhibitor B43-Genistein	Dose level 0.1 mg/kg
**Other agents**									
[61]	Baluna et al., 1996	NA	USA	56	12	21.4	Non-Hodgkin lymphoma	Ricin A chain-containing immunotoxin	Using IgG-HD37-RTA continuous infusion 9.6–19.2 mg/m^2^(2), bolus infusion range 2–24 mg/m^2^(2) IgG-RFB4-RTA continuous infusion 9.6–28.8 mg/m^2^(4), bolus infusion 23–48 mg/m^2^(2) Fab’-RFB4-RTA bolus infusion 25–100 mg/m^2^(2)
[62]	Borghaei et al., 2009	NA	USA	39	6	15.4	NSCLC, pancreatic cancer	ABR-217620	Dose level 20 μg/kg
[63]	Hochhauser et al., 2009	2004–2006	UK	16	10	62.5	Ampulla of vater cancer, cholangiocarcinoma, colorectal cancer, lung cancer, esophagus cancer, pancreatic cancer, sarcoma, malignant melanoma, stomach cancer	Pyrrolobenzodiazepine	Using dose 15–240 μg/m^2^
[64]	Posey et al., 2002	NA	USA	46	1	2.2	Colorectal cancer, pancreatic cancer, ovarian cancer, breast cancer, lung cancer, prostate cancer, head and neck cancer, stomach cancer, endometrial cancer, thyroid cancer, unknown primary lesion	SGN-10 (or BR96 sFv-PE40)	Dose level > or = 0.384 mg/m^2^
[65]	Elias et al., 2001	NA	USA	5	4	80.0	Breast cancer	Paclitaxel	Dose level 150 mg/m^2^
[66]	Grossbard et al., 1993	1990–1991	USA	12	5	41.7	Non-Hodgkin lymphoma	Anti-B4-bR	Using dose 20, 40, 50 μg/kg/day for 7 days MTD 40 μg/kg
[67]	Pazdur et al., 1991	NA	USA	17	6	35.3	Metastatic cancer	FK973	Using dose 30 mg/m^2^(2), 45 mg/m^2^(4)
[68]	Barrett et al., 1982	1980–1981	UK	36	4	11.1	Acute myeloid leukemia, acute lymphoblastic leukemia, aplastic anemia, mucopolysaccharidosis, metachromic leukodystrophy	Dihydro benzoxazine	Using dose 12.5 mg/kg(10), 500 g/m^2^(26)
[69]	Zwaan et al., 2014	NA	Multicenter in Europe†	36	3	8.3	Acute myeloid leukemia	Cyclosporine	Using dose plasma concentration <100 μg/L
[69]	Zwaan et al., 2014	NA	Multicenter in Europe†	29	1	3.4	Acute myeloid leukemia	Clofarabine + cytarabine + liposomal daunorubicin	Clofarabine 20, 30, 40 mg/m^2^ Ara-C 2 g/m^2^/day dauorubicin 40–60 mg/m^2^

NA: not available (information was not included in the case series article), CLS: capillary leak syndrome, Using dose: drug dose that was administered to patients, Dose level: serum drug level when the patients show toxicity, DLT: dose limited toxicity, IL: Interleukin, *w*/*v*: weight/volume percentage, ch14.18: a chimeric human/murine anti-GD2 antibody, MIU: million international units, GVHD: graft-versus-host disease, INF: interferon, GM-CSF: granulocyte-macrophage colony-stimulating factor, G-CSF: granulocyte-colony stimulating factor, 5-FU: 5-fluorouracil, SS1P: recombinant anti-mesothelin immunotoxin, CD: cluster of differentiation, MTD: maximum tolerated dose, NSCLC: Non small cell lung cancer, ABR-217620: naptumomab estafenatox, SGN-10: a single-chain immunotoxin, Anti-B4-bR: B-cell restricted immunotoxin anti-B4-blocked ricin, FK973: novel, substituted dihydro benzoxazine structurally similar to mitomycin, USA: United States of America, UK: United Kindom; *All study patients developed capillary leak syndrome after receiving anti-cancer agents. There were no capillary leak syndrome features before treatment. † Study population was collected from multiple centers in Europe: Netherlands, Austria, Germany, France, the Czech Republic, and the United Kingdom.

**Table 2 jcm-08-00143-t002:** Summary profiles of clinical studies that reported capillary leak syndrome related to bone marrow transplantation.

Ref. No.	Author, Year	Total Number	CLS	Incidence (%)	Diagnosis	Hypothesis or Risk Factors
Only BMT related						
[70]	Cahill, et al., 1996	55	29	52.7	Both allogeneic and autologous transplant recipients	Pivotal contribution by circulating leukocytes
[71]	Nurnberger, et al., 1993	12	4	33.3	Acute lymphoblastic leukemia Aplastic anemia Fanconi’s anemia Neuroblastoma Ewing’s sarcoma Lymphoepithelial carcinoma	C1 Inhibitor activity decreased to 0.60-fold to 0.80-fold Elevated C4d concentrations (up to 2.4 mg/dL, upper normal threshold value: 0.9)
[72]	Nurnberger, et al., 1997	96	20	20.8	Acute lymphoblastic leukemia Acute myeloblastic leukemia Chronic myeloblastic leukemia Severe aplastic anemia Ewing tumors Rhabdomyosarcoma Neuroblastoma Lymphoepithelioma	Receiving G-CSF or GM-CSF* GVHD prophylaxis : MTX plus cyclosporine A Allogeneic-related BMT, solid tumor Unrelated BMT, hematologic disease Patients with high-risk pretreatment
[40]	Gorin et al., 1992	44	3	6.8	Non-Hodgkin’s lymphoma	BMT after using GM-CSF* ( Dose level 250 μg/m^2^)
[42]	Steward et al., 1989	20	3	15.0	Metastatic solid tumors	BMT after using GM-CSF* (Using dose 0.3, 1.0, 3.0, 10, 30, and 60 μg/kg/day, dose level 32 μg/kg)
[72]	Nurnberger, et al., 1997	142	22	15.5	Acute lymphoblastic leukemia Chronic myelomonocytic leukemia Severe aplastic anemia Fanconi’s anemia Non-Hodgkin’s lymphoma, Ewing tumor Neuroblastoma Rhabdomyosarcoma Wiskott–Aldrich syndrome	BMT after using G-CSF* Low levels of C1 esterase inhibitor†
[73]	Salat et al, 1995	48	7	14.6	Acute lymphoblastic leukemia Acute myeloblastic leukemia Chronic myeloblastic leukemia Hodgkin’s lymphoma Non-Hodgkin’s lymphoma Severe aplastic anemia Multiple myeloma	Elevation of terminal complement complex (TCC) levels Elevation of functional Cl-esterase inhibitor (CI-INH) Elevation of Cl-inhibitor antigen (CI-INH antigen)

CLS: capillary leak syndome, GM-CSF: granulocyte-macrophage colony-stimulating factor, G-CSF: granulocyte-colony stimulating factor, GVHD: Graft-versus-host disease, MTX: methotrexate, BMT: bone marrow transplantation; * These patients initially received bone marrow transplantation, and then received GM-CSF to correct neutropenia; † To correct this status, 15 severe CLS patients were treated with C1 INH concentrate using a cumulative dose of 180 units/kg in this article.

**Table 3 jcm-08-00143-t003:** Meta-analyses on the incidence of capillary leak syndrome induced by various anti-cancer drugs or after BMT in cancer patients.

Causative Drugs	Numberof Studies	Total Number of Patients	Number of CLS	Incidence of CLS (Overall)	Incidence of CLS by Meta-Analysis (95%CI)	Heterogeneity I^2^ (*p* Value)	Incidence of CLS Median (Ranges)
IL-2	18	703	244	34.7%	43.9% (29.5–58.9)	92.6% (*p* < 0.0001)	32.4% (5.3–100)
IL-2 with other agents	13	405	118	29.1%	32.0% (15.6–51.1)	91.1% (*p* < 0.0001)	16.7% (0–100)
IL-2 + IFN-alpha 2a	2	55	47	85.5%	90.4% (64.1–100)	80.0% (*p* = 0.0255)	90.3% (80.5–100)
IL-2 + imatinib mesylate	3	17	2	11.8%	15.0% (3.1–33.4)	0% (*p* = 0.4889)	9.0% (0–33.3)
IL-2 + bevacizumab	1	4	4	100.0%	-	-	-
IL-2 + 5-FU	2	40	7	17.5%	17.1% (3.7–37.4)	56.1% (*p* = 0.1312)	33.3% (6.3–25.0)
IL-1 with other agents	2	24	10	41.7%	42.3% (24.3–61.4)	0% (*p* = 0.8266)	42.2% (40–44.4)
IL-4 (+IL-2)	1	17	2	11.8%	-	-	-
GM-CSF	3	78	7	9.0%	10.1% (4.6–17.6)	0% (*p* = 0.5802)	7.1% (6.8–15.0)
Gemcitabine	3	86	3	3.5%	4.9% (1.4–10.3)	0% (*p* = 0.9273)	3.7% (2.8–4.3)
SS1P	2	58	15	25.9%	26.9 (0.00–78.6)	94.5% (*p* < 0.0001)	30.1 (5.9–54.2)
Anti-CD agents	13	221	75	33.9%	35.6% (16.1–60.0)	91.8% (*p* < 0.0001)	20.0% (5.9–100)
Anti-CD22	4	59	24	40.7%	48.1% (6.3–91.7)	93.7 (*p* < 0.0001)	44.1% (11.5–100)
Anti-CD19 + anti-CD22	2	42	8	19.0%	17.8% (2.7–42.2)	69.6% (*p* = 0.0699)	17.0% (5.9–28.0)
Anti-CD25	3	60	22	36.7%	42.2% (0.02–98.0)	97.0% (*p* < 0.0001)	11.1% (6.7–100)
BMT	7	417	88	21.1%	21.7% (12.2–33.1)	83.9% (*p* < 0.0001)	15.5% (6.8–52.7)
Only BMT-related	3	163	53	32.5%	35.5% (14.7–59.6)	87.5% (*p* = 0.0003)	33.3% (20.8–52.7)
BMT with other agents	4	254	35	13.8%	14.2% (10.2–18.7)	0% (*p* = 0.5001)	14.8% (6.8–15.5)

CLS: capillary leak syndome, IL: interleukin, GM-CSF: granulocyte-macrophage colony-stimulating factor, 5-FU: 5-fluorouracil, SS1P: recombinant anti-mesothelin immunotoxin, CD: cluster of differentiation, BMT: bone marrow transplant.

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
