# Peer review of "Incidence of Capillary Leak Syndrome as an Adverse Effect of Drugs in Cancer Patients: A Systematic Review and Meta-Analysis"

_jcm, 2019, doi:10.3390/jcm8020143_

Round 1
Reviewer 1 Report
This study is the first to provide an informative overview of the incidence rate
of reported CLS patients as an adverse event of cancer treatment. This meta-analysis should lead to
a better understanding of CLS and assist physicians in identifying the presence of CLS early in the
disease course to improve the outcome and optimize management. However I have some concerns which are detailed below:
1. Please change the quality of the figure.
2. Some font sizes are not consistent all over the text.
3. delete "the" line 72 ....from capillaries, resulting in the clinical features
4. add "the high" line 75 of the nonspecific clinical features and high mortality rate of CLS [2].
5. add comma after hemoconcentration(hypoalbuminemia, hemoconcentration and hypotension).
6. delete comma ..The incidence of CLS associated with BMT with or without other agents, ranged ..
from 6.8 to 52.7%.
7. add "a" dose-dependent anti-CD22 mAb (4 studies) showed dose-dependent.
8. change to define to "defining"...placebo-controlled trials with a control arm that would allow to
define how...
9. add "the incidence"...Incidence of CLS due to IL-2 (18 studies) was 34.7% by overall.
10. add "a choice or the choice " risk of development of CLS in choice of treatment and preparation of appropriate management for cancer patients in anticipation of this s
Author Response
Response to Reviewer 1 Comments
Point 1: Please change the quality of the figure.
Response 1: We thanks for your observations. We have made the suggested changes.
Point 2:Some font sizes are not consistent all over the text.
Response 2: We have found that the caption below Figure 2 is incorrectly included in the manuscript, so we have revised it in the manuscript.
Point 3:delete "the" line 72 ....from capillaries, resulting in the clinical features
Response 3:We have deteted “the” in the manuscript.
Point 4:add "the high" line 75 of the nonspecific clinical features and high mortality rate of CLS [2].
Response 4:We have added “the high” in the manuscript.
Point 5:add comma after hemoconcentration (hypoalbuminemia, hemoconcentration and hypotension).
Response 5:We have added comma in the manuscript.
Point 6:delete comma ..The incidence of CLS associated with BMT with or without other agents, ranged ..from 6.8 to 52.7%.
Response 6:We have deleted comma in the manuscript.
Point 7: add "a" dose-dependent anti-CD22 mAb (4 studies) showed dose-dependent.
Response 7:We have added “a” in the manuscript.
Point 8:change to define to "defining"...placebo-controlled trials with a control arm that would allow to define how...
Response 8:We have replaced “to define” to “defining” in the manuscript.
Point 9:add "the incidence"...Incidence of CLS due to IL-2 (18 studies) was 34.7% by overall.
Response 9:We have added “the” in the manuscript.
Point 10:add "a choice or the choice " risk of development of CLS in choice of treatment and preparation of appropriate management for cancer patients in anticipation of this s
Response 10:We have added “a” in the manuscript.
Reviewer 2 Report
This review reports on the prevalence of capillary leak syndrome (CLS) associated with and attributed to cancer chemotherapy. Unfortunately, this report is nearly identical to the authors’ own (Ref 4, which largely cites the same studies) and other recent reviews (PMID: 30244023) and adds nothing to the published literature. Most studies cited were published prior to 2010, making this review of dubious relevance to modern chemotherapy.
What’s more, the authors have provided no criteria for the diagnosis of CLS, instead lumping drug-related CLS together with “Clarkson syndrome”, which is by definition a distinct disorder of unknown etiology, different prognosis and treatment, and unrelated to secondary causes of CLS, such as a chemotherapy or BMT. This is explicit misinformation.
Other comments:
In the discussion, the authors state that CLS is due to “endothelial damage”. What is the evidence for this?
Author Response
Response to Reviewer 2 Comments
Point 1: Unfortunately, this report is nearly identical to the authors’ own (Ref 4, which largely cites the same studies) and other recent reviews (PMID: 30244023) and adds nothing to the published literature. Most studies cited were published prior to 2010, making this review of dubious relevance to modern chemotherapy.
Response 1: Thank you for the very insightful comments. The article in Ref 4 is our previous systematic review of sporadic case reports with capillary leak syndrome (CLS) as a presenting symptom of cancers or cancer drug-related CLS. The aim of that study was to evaluate the clinical characteristics, laboratory findings, clinical course, treatment patterns of CLS and mortality rate in cancer patients with CLS, because there has been no report on these issues (how about the clinical symptoms, course and outcome and how should we treat the patients with CLS associated with cancers or cancer drug-related CLS). However, this study cannot answer about the incidence of CLS for each cancer drug, because the case reports were reported sporadically in the literature. In contrast, CLS has been reported as one of the adverse events in many hemato-oncology clinical trials despite no detailed informations on the clinical course and outcome. Nevertheless, we can calculate the incidence of CLS for each cancer drug. Therefore, our submitted article was conducted to answer about the question of how about the incidence of CLS for each cancer drug (how many or how about the percentage of CLS develops for each cancer drug at the dose used) and knowing this epidemiologic incidence for CLS is very important in the treatment of cancer patients. Because our study summarizes the number of CLS per total patients treated with anti-cancer drug as a side effect reported in clinical trials of cancer patients, focusing on the incidence of the disease, individual clinical trials cited and included in our article are totally different (not same study) from our previous systematic review of scattered case reports which are not reported as a result of clinical trials. In addition, we also estimated and performed meta-analysis on the incidence rate of CLS after bone marrow transplantation (BMT) which has not been reported so far.
According to the comment, we also checked and reviewed the recently published article (PMID: 30244023) and found that it was on the cumulative number of patients of CLS as a side effect of different type of drugs and it was results of database in which if a CLS develops, the case is registered in this database. Therefore, the database is a just collection of sporadically reported cases, but the incidence rate of CLS for cancer patients treated with drugs cannot be calculated because the database is there is no report on the total patients.
Thus, our submitted article on meta-analysis of incidence for CLS by anti-cancer drug or after BMT is the first in the literature and the two studies (our previous systematic review and PMID: 30244023) are different from each other and have different messages.
Point 2: the authors have provided no criteria for the diagnosis of CLS, instead lumping drug-related CLS together with “Clarkson syndrome”, which is by definition a distinct disorder of unknown etiology, different prognosis and treatment, and unrelated to secondary causes of CLS, such as a chemotherapy or BMT. This is explicit misinformation.
Response 2: We apologize for unintentionally being misleading and thank you for your insightful comments. As you mentioned, capillary leak phenomenon may be similar between idiopathic CLS and secondary CLS due to drugs or after BMT, but the pathophysiology of them can be somewhat different. For example, CLS by anti-cancer drugs could also develop due to direct toxicity to the capillary system. Also, the exact definition of capillary leak has not been established yet. According to the comments, we additionally described these points of discussion to the manuscript.
Point 3: In the discussion, the authors state that CLS is due to “endothelial damage”. What is the evidence for this?
Response 3: Thank you for your insightful comment. The episodes of capillary leak syndrome such as hypotension, hemoconcentration, and hypoalbuminemia are thought to be caused by endothelial dysfunction. Our study has been focused on the drug-induced capillary leak syndrome associated with drugs such as Interleukin-2, which are known to cause endothelial damage. Therefore, we have mentioned the association of these pathogenic drugs with endothelial damage, not with capillary leak syndrome. The phrases that can cause misunderstandings have been modified according to your suggestion.